# Precision Oncology of High-Grade Ovarian Cancer Defined through Targeted Sequencing

**DOI:** 10.3390/cancers13205240

**Published:** 2021-10-19

**Authors:** Sandra Wessman, Beatriz Bohorquez Fuentes, Therese Törngren, Anders Kvist, Georgia Kokaraki, Hanna Menkens, Elisabet Hjerpe, Ythalo Hugo, Tirzah Braz Petta, Åke Borg, Joseph W. Carlson

**Affiliations:** 1Department of Oncology-Pathology, Karolinska Institutet, SE-171 76 Stockholm, Sweden; sandra.wessman@ki.se (S.W.); beatriz0815@gmail.com (B.B.F.); georgia.kokaraki@ki.se (G.K.); hanna.menkens@gmail.com (H.M.); 2Department of Clinical Sciences, Lund University, SE-223 81 Lund, Sweden; therese.torngren@med.lu.se (T.T.); anders.kvist@med.lu.se (A.K.); ake.borg@med.lu.se (Å.B.); 3Department of Pathology and Laboratory Medicine, Keck School of Medicine, University of Southern California, Los Angeles, CA 90089, USA; pesquisaclinica.tirzah@gmail.com; 4Visby Lasarett, SE-621 55 Visby, Sweden; elisabet.hjerpe@gotland.se; 5Institute of Education, Research and Invention, Liga Contra o Câncer, Natal 59075-740, RN, Brazil; ythalo.hugo@liga.org.br; 6Department of Cellular Biology and Genetics, Federal University of Rio Grande do Norte, Natal 59075-740, RN, Brazil

**Keywords:** ovarian cancer, genomics, genetics, targeted therapy

## Abstract

**Simple Summary:**

Ovarian cancer is a rare and deadly gynaecologic cancer, with a relatively large hereditary component. Genomic analysis of tumour material can potentially provide information regarding therapy and identify hereditary carriers and their families. The aim of our prospective study was to apply genomic characterization to tumour material from women with ovarian cancer to identify women that might benefit from PARP inhibitor therapy, as well as to detect and triage women to genetic counselling. We used next generation sequencing using a targeted panel to prospectively analyse 274 tumours and identified 50 with a BRCA1/2 pathogenic variant. Twenty patients received olaparib based on these results, and 16 previously unknown hereditary carriers were identified. In addition, in a subset examined by an extended sequencing panel, actionable mutations were found in 84/88 tumours. This study demonstrates that personalized medicine approaches can be useful for women with ovarian cancer and can help with therapy selection and identification of at risk families.

**Abstract:**

Background: We examined whether molecular characterization of high-grade epithelial ovarian cancer can inform the diagnosis and/or identify potential actionable targets. Methods: All of the consecutively sequenced high-grade ovarian tumours with consent between 2014 until 2019 were included. A total of 274 tumours underwent next generation sequencing using a targeted panel. Results: Patients with high-grade ovarian epithelial cancer were consented to prospective molecular characterization. Clinical information was extracted from their medical record. Tumour DNA was subjected to sequencing, and selected patients received PARP inhibitor therapy. Conclusions: Tumours from 274 women were sequenced, including high-grade serous carcinoma (*n* = 252), clear cell carcinoma (*n* = 4), carcinosarcoma (*n* = 9), endometrioid carcinoma (*n* = 3), undifferentiated carcinoma (*n* = 1), and mixed tumours (*n* = 5). Genomic profiling did not influence histologic diagnosis. Mutations were identified in *TP53*, *BRCA1*, *BRCA2*, as well as additional homologous recombination repair pathway genes *BARD1*, *ATR*, *CHEK2*, *PALB2*, *RAD51D*, *RAD50*, *SLX4*, *FANCA*, *RAD51C*, and *RAD54L*. In addition, mutations in *PTEN* and *CDKN2A* were identified. Several somatic mutations with implications for germline testing were identified, including *RMI1*, *STK11*, and *CDH1*. Germline testing identified 16 previously unknown BRCA1/2 carriers. Finally, 20 patients were treated with the PARP inhibitor olaparib based on the sequencing results.

## 1. Introduction

Ovarian cancer is a rare and deadly gynaecologic cancer, killing approximately 180,000 women in the world every year [1]. It is considered an orphan disease by the FDA, and thus therapies for ovarian cancer can receive special status during the approval process [2]. The behaviour and biology of ovarian cancer is closely related to the histotype of the tumour, which is established through histologic appearance, including cytologic atypia and mitotic rate, and the selected use of immunohistochemistry [3,4]. The most common histologic subtype of ovarian cancer is high-grade serous, but other high-grade subtypes, such as clear cell carcinoma, endometrioid, and carcinosarcoma also contribute to ovarian cancer mortality [5]. Ovarian cancer has a strong hereditary component, with approximately 10% of cases being related to Hereditary Breast and Ovarian Cancer Syndrome. HBOC, in turn, is related to inherited variants in a number of genes, particularly *BRCA1* and *BRCA2*, but also secondary genes such as *RAD51C* or even potentially other, as yet not fully described, genes [6]. Molecular characterization of ovarian cancer cases has been increasingly used to improve diagnosis and guide therapy, as well as to potentially identify hereditary carriers for cascade genetic testing.

To determine whether genomic profiling could improve diagnosis, assist with prognosis or aid in therapy selection, we prospectively characterized ovarian cancers with a focus upon high-grade subtypes. We applied a clinically validated NGS platform to determine whether molecular profiling could improve classification and subtyping, provide additional prognostic information, identify actionable mutations, and contribute to the identification of patients with hereditary risk. Moreover, we evaluated whether patients with the actionable mutations received therapy, and, if so, what benefit they may have derived.

## 2. Materials and Methods

### 2.1. Patient Inclusion

Patients undergoing surgery for ovarian cancer at the Karolinska University Hospital were consented to prospective molecular testing using NGS under a protocol approved by the ethical review board. Age is reported based upon age at diagnosis.

### 2.2. Histologic Diagnosis, Stage, and Grade

All of the consecutively sequenced high-grade ovarian tumours with consent between 2014 until 2019 were included. Histologic subtypes considered “high-grade” were high-grade serous, carcinosarcoma, clear cell, undifferentiated, and endometrioid (FIGO grade 3). All of the tumours were diagnosed by the gynaecological pathology group at the Karolinska University Hospital. Diagnosis of histotype and grade were assigned using the established criteria, and relied on the evaluation of the haematoxylin and eosin stained slides, as well as immunohistochemical markers, including CK7, CK20, PAX8, ER, PR, WT-1, and p53 [7]. Tumours were staged according to the International Federation of Gynaecology and Obstetrics (FIGO) staging system [7].

### 2.3. Genomic Sequencing

Sequencing was performed at the BRCA Lab at Lund University, Department of Clinical Sciences, Lund, Sweden. For the purposes of this paper, “clinical sequencing” refers to sequencing performed for a clinical indication, in a clinically licensed reference lab. Sequencing was performed on either fresh frozen tissue (*n* = 89) or, if fresh tissue was unavailable, on FFPE material (*n* = 185). The tumour content of the fresh frozen material was confirmed by the pathologist using a paired FFPE block. A minimum of 30% tumour cell proportion was required. From fresh frozen tissue genomic DNA was enriched by hybrid capture using a custom panel. This included the BRCA1 and BRCA2 genes, as well as an extended panel of 62 additional genes related to breast and ovarian cancer. The hybrid capture probes cover 99.97% of coding regions and 20 bp adjacent introns of targeted genes (Appendix A).

The resulting DNA was sequenced (Illumina) to a minimum of 100× coverage of coding exon +/− 20 bp adjacent intron sequence. Alignment was performed using Novoalign, variant identification was performed using GATK, Freebays, Manta, XHMM (providing copy number variation) and Melt. For FFPE tissue genomic DNA, one of two methods was used to enrich for *BRCA1* and *BRCA2*: Single molecule molecular inversion probes (smMIPs, [8]) or TruSeq Custom Amplicon (v.15, Illumina, San Diego, CA, USA) with complementary amplicons sets to target both DNA strands. Both panels amplify 100% of coding regions and 20 bp adjacent introns of *BRCA1* and *BRCA2* genes (Appendix A).

Amplified libraries were sequenced (Illumina) to a minimum of 100× coverage of coding exon +/− 20 bp adjacent intron sequence. Alignment and variant identification were performed using SeqPilot v4.4.0, Module: SeqNext, JSI for smMIPs and MiSeq Reporter (v2.5, Illumina) for TruSeq Custom Amplicon libraries. Detected variants were visualized in the Integrative Genomics Viewer (IGV, [9]) and supporting sequences were reviewed in genomic context to identify and exclude artefacts. A majority of the reported *BRCA1* and *BRCA2* variants were confirmed using Sanger sequencing (70%), smMIPs (9.4%, only variants detected by hybrid capture) or pyrosequencing (9.4%), with a 100% success rate. A minority of *TP53* variants were also confirmed by Sanger sequencing (two variants) or smMIPs (11 variants) with a 100% success rate. Variants in other genes were reviewed in IGV. A minimum of 5% allele rate was required for the identified variants. Variant pathogenicity was determined using ClinVar [10]. All of the variants described are pathogenic/likely pathogenic based upon the clinical significance and class from ClinVar, unless specifically noted in the text (i.e., in the case of *BRCA1/2* variants of uncertain significance). 

### 2.4. Annotation of Somatic Variants

To classify the actionability and level of evidence of individual somatic genomic variants, we utilized the OncoKB knowledge database [11]. These annotations were performed as of 15 March 2021. A comparison was made to the Cancer Genome Atlas Ovarian Serous Cystadenocarcinoma cohort using the cBioPortal for cancer genomics as of 15 March 2021 [12,13,14]. *TP53* alterations were assessed using the IARC TP53 database (R20, July 2019 version) [15]. Of note, at the time of this study, the indication in Sweden for the treatment with olaparib was a demonstrated *BRCA1/2* P/LP variant identified either as germline or somatic, and a response to second (or more) line platinum therapy. In Sweden, at the time of this study, only selected ovarian cancer patients were referred to clinical genetics. This is in contrast to many countries where all of the women with ovarian cancer were referred. According to the guidelines, this was to be done in cases with diagnosis at young age and/or a medical history of breast cancer and/or at least one first degree relative with breast- or ovarian cancer. All of the patients with a detected tumour *BRCA*-mutation were referred to clinical genetics. 

## 3. Results

### 3.1. Patient Demographics

Tumour material from a total of 274 patients was successfully sequenced. The median age of the patients was 66 years (range 34–89). Unless otherwise specifically noted, all of the *BRCA1/2* variants described are pathogenic/likely pathogenic. The median age among germline *BRCA1/2* carriers was 59 years compared to the median age of 65 years among patients with somatic *BRCA1/2*-variants and 67 years among patients negative for *BRCA1/2* variants (*p* = 0.02; Table 1). Clinical characteristics of the cohort are summarized in Figure 1. Most tumours (91.9%) were of high-grade serous histotype, with other high-grade subtypes only rarely represented (Table 1). Most tumours (83.8%) were stage III or IV, representing a spread outside the ovary at the time of surgery. There was no difference in stage for germline *BRCA1/2* carriers (Table 1). The median follow-up time was 21.6 months (1.2–117.6) and at the last follow-up 94 patients had no evidence of disease (NED), 111 were alive with disease (AWD), and 69 had died of disease (DOD). 

### 3.2. Genomic Alterations, Histotype, and Prognosis

Most of the tumours analysed were classified as high-grade serous. The extended sequencing panel was applied to 81 of these, of which 75/81 (93%) showed a *TP53* variant. This number is similar to the 88% of samples with *TP53* mutations identified in high-grade serous ovarian cancer cohort in the TCGA (Appendix A) [14]. Two of the high-grade serous cases showed *PTEN* variants, and these have been described previously in high-grade serous carcinomas. Only one clear cell carcinoma underwent extended sequencing, and this case showed both *PTEN* and *BARD1* mutations. Four carcinosarcomas underwent extended sequencing, and all four demonstrated *TP53* mutations, as expected. Finally, one endometrioid tumour underwent extended sequencing, with no mutations identified, and one mixed tumour, with a *TP53* variant. Therefore, extended sequencing did not appear to affect diagnostic histotype or tumour grade.

### 3.3. Genomic Alterations and Therapeutic Actionability

The tumour material from all of the 274 patients underwent sequencing for the *BRCA1* and *BRCA2* genes. In 88 patients, the extended sequencing panel was performed. In 59/274 (21.5%) of patients, a *BRCA1/2* variant was detected, including 50 pathogenic or likely pathogenic (P/LP, Class 4 or 5), and nine variants of unknown significance (VUS, Class 3, Appendix A). 

Of the 50/274 (18%) cases with P/LP variants, 29 were detected within *BRCA1* (TCGA 6%) and 21 within *BRCA2* (TCGA 4%, Appendix A). All of the detected *BRCA1* P/LP variants were seen in patients with high-grade serous carcinoma. One of the *BRCA2* P/LP variants was seen in a patient with a mixed histologic subtype. The *BRCA* variants with the associated clinicopathologic data are presented in Table 2. The mutations were distributed across the length of both the *BRCA1* and *BRCA2* genes. Of note, we additionally observed a high prevalence of the Swedish founder mutation *BRCA1* 3171ins5 (4/37; 10.8%) [14]. 

The extended sequencing panel was used to perform a genomic analysis for additional actionable mutations (summarized in Figure 2A). Among actionable targets, homologous recombination repair (HRR)-related gene analysis revealed tumours that could be potentially targeted with PARP inhibitors [16,17]. The first, and most frequent, non-*BRCA1/2* alteration seen was in *BARD1*, detected in four tumours. The TCGA high-grade serous ovarian cancer cohort showed two cases with missense mutations of unknown significance. Germline alterations in this gene have been linked to an increased risk of breast and ovarian cancer [18,19]. Moreover, one of these patients had a tumour with a *CDKN2A* alteration (gene discussed below), and a second patient had a concomitant alteration in *BLM*. The BARD1-BRCA1 complex is involved in various aspects of DNA repair, gene expression, replication fork maintenance, and chromatin regulation [20,21]. 

Another HRR-related gene is *ATR*, and alterations in this gene were seen in two tumours. Somatic mutations are associated with microsatellite instability and are also found in colon cancer, urothelial cancer, gastric cancer, endometrial cancer, and myelomas [22]. The *ATR* inhibitor M6620 (VX-970) is currently in phase I clinical trial targeting solid tumours with defective *ATM* signalling [22].

Moreover, the *CHEK2* gene is HRR-related and was altered in one tumour (with a concomitant *BRCA1* alteration). Checkpoint Kinase 2 (CHEK2) is a serine/threonine kinase that plays a central role in the DNA damage checkpoint pathway. CHEK2 is activated by the kinases ATR and ATM via phosphorylation [22].

Another HRR-related gene, *PALB2*, was altered in a single tumour. Moreover, the TCGA cohort showed rare alterations in the gene (Table 1). *PALB2* acts as a support protein in the homologous recombination (HR) pathway for the repair of double-stranded DNA breaks. It likely mediates recruitment of RAD51 and BRCA2 at the damaged loci [23]. Three tumours showed alterations in *RAD51D*, one of which showed a concomitant P/LP variant in *BRCA2*. The TCGA cohort contained four tumours with *RAD51D* mutations, corresponding to a frequency of 1.3%. One tumour showed alterations in both *RAD50* and *SLX4*. Both of these alterations can be rarely found in the TCGA cohort, but not concomitantly. RAD50 is a subunit of the MRE11/RAD50/NBS1 (MRN) complex [24]. The SLX4 protein is involved in various processes related to DNA damage repair [25]. One tumour showed an alteration in *FANCA*, one tumour showed an alteration in *FANCM* (with a concomitant *BRCA1* P/LP variant), and one tumour showed an alteration in *RAD51C* [26]. In addition, one tumour showed an alteration in *RAD54L* [27]. All of the above alterations in HRR-related genes may indicate that these tumours could be sensitive to PARP inhibitors such as niraparib, olaparib, and rucaparib. Moreover, there is good evidence from prostate cancer that the PARPi olaparib is effective in tumours with alterations in *RAD51C, RAD51D, CHEK2, PALB2, RAD54L*, and *BARD1*. In summary, a total of 33/88 (38%) of tumours showed an alteration in an HRR-related gene, of which 19 were *BRCA1/2* and 14 were additional genes.

The extended panel revealed an alteration in *TP53* in 80/88 (91%) of patients (Figure 2B). This is similar to the frequency reported in the TCGA serous carcinoma cohort of 88%. *TP53* mutation subtype analysis revealed a total of 48 individual mutations. For the 30 missense mutations, SIFT analysis classified all of the mutations as damaging, and the transactivation class showed only one mutation type (seen in only one patient) as “functional”. Hotspot mutations were detected in tumours from 21 patients [28]. Adavosertib, an inhibitor of the nuclear kinase *WEE1*, has emerged as a promising candidate for the sensitization of *TP53* mutated tumour cells to chemotherapy. The recently reported double blind phase 2 trial indicated that the class of *TP53* mutation may be useful as a potential response biomarker, with hotspot mutations showing the strongest response [29]. 

The extended sequencing panel identified additional gene alterations that may be actionable. Four tumours showed alterations in *PTEN*. One of these tumours had a concomitant alteration in BARD1, and a second in *BRCA1* and *RMI1*. Class 4 evidence exists for the drugs GSK2636771 and AZD8186. *PTEN* is a tumour suppressor that is one of the most frequently mutated genes in human cancer [30]. One tumour had a large (>130 kb) deletion on chromosome 9p21 affecting several genes including tumour suppressor *CDKN2A*, which encodes both the CDK4/6 interacting p16-IN4A protein, but also the p14-ARF protein that has a function in the MDM2-TP53 pathway, thus it is relevant in ovarian tumours. There is a level 4 evidence for the potential use of abemaciclib, palbociclib, and ribociclib in *CDKN2A* mutated tumours. 

Taken as a whole, only four patients lacked an actionable target after extended panel sequencing. The 84 patients with an actionable target included *TP53* mutated tumours (79), HRR-related gene alterations (19 tumours), and other actionable targets (*PTEN* and *CDKN2A*, five tumours).

Finally, three tumours showed alterations that are not currently actionable for therapy, but may have implications for germline testing. One tumour showed an alteration in RecQ Mediated Genome Instability 1 (*RMI1*, associated with Bloom Syndrome and Baller-Gerold Syndrome). One tumour showed an alteration in *STK11*. The alteration in this gene was described in 16 cases in the TCGA cohort (5%). Germline alterations in this gene are seen in Peutz-Jeghers syndrome. One tumour showed an alteration in *CDH1* (TCGA 5%). Germline alterations in *CDH1* are seen in Hereditary Diffuse Gastric Cancer. 

Somatic P/LP variants in *BRCA1* and *BRCA2* are actionable with PARP inhibitors niraparib, olaparib, and rucaparib. Under Swedish guidelines, at the time of this study, patients were evaluated for maintenance therapy with olaparib. Of the 50 patients with P/LP variants in *BRCA1/2*, a total of 17 received olaparib. Moreover, an additional three patients without demonstrated *BRCA1/2* P/LP variants received olaparib. The follow-up of the patients that were treated with olaparib is presented in Figure 3. 

### 3.4. Genomic Alterations and Germline Testing

A total of 94 patients were referred for genetic counselling, including 48 patients with a tumour detected *BRCA1/2* alteration. Of these 48 patients, 26 were P/LP *BRCA1* variants, 19 were P/LP *BRCA2* variants, and three were VUS. After germline sequencing, 16 *BRCA1* germline carriers were identified and 13 *BRCA2*. Of the 29 germline carriers identified, 16 had a relevant family cancer history.

The extended gene panel identified two potentially hereditary alterations. First, a *RAD51D* deletion that most likely is a germline (Class 4) variant, since it has been observed repeatedly (>5×) in oncogenetic screening of breast ovarian cancer families. Second, a *RAD51C* alteration that was confirmed separately through germline testing.

## 4. Discussion

Ovarian cancer is a collective name for a heterogeneous group of tumours with diverse clinical and molecular characteristics. Epithelial ovarian cancer, the focus of this project, is now understood to consist of five primary tumour histotypes: High-grade serous, clear cell, endometrioid, mucinous, and low grade serous [3,4]. The first three can show diagnostic overlap, including immunohistochemical markers. They are often characterized by homologous recombination related gene defects, and high levels of copy number variation. Clinical sequencing has primarily focused on therapy with PARP inhibitors or has been used in clinical trials to evaluate one specific gene [31].

Findings of clinical utility included confirmation of mutations in *TP53* for more accurate diagnosis, and detection of potentially actionable mutations in homologous recombination related genes, *ATR*, *CHEK2*, *TP53*, *PTEN*, and *CDKN2A*. These findings could lead to the selection of FDA-approved therapies or the identification of patients eligible for clinical trials. Molecular characterization identified a relatively large number of high-grade serous tumours that lacked a *TP53* mutation. The lack of *TP53* mutation has potential therapeutic consequences and may even have other prognostic or clinical implications. Deeper analysis revealed that 21/88 patients had hot spot *TP53* mutations. The recent clinical trial of adavosartib demonstrated that patients with hotspot mutations had the greatest response to therapy. Twenty patients received olaparib during the study period, of which 17 had a demonstrated *BRCA1/2* mutation. PARP inhibitors are becoming more widespread as a therapy for HRR-gene mutated tumours. Three recent clinical trials have demonstrated enhanced disease-free survival in selected patients [32,33,34]. NGS extended panel sequencing identified an additional 19/89 patients with non-*BRCA1/2* HRR-gene mutations. 

In total, actionable mutations were identified in 84/88 patients that underwent extended sequencing, and 20 patients received olaparib.

Our data identified four patients with *BARD1* mutations. BARD1 interacts with BRCA1 via its N-terminal RING domain, and pathogenic variants of *BRCA1* disrupt this interaction [18,21]. Therefore, it has been hypothesized that variants in *BARD1* could also influence the function of *BRCA1*. Interestingly, all of the *BARD1* mutations detected in the cohort were associated with mutations in a second tumour suppressor (three with *TP53*, one with PTEN). Within the TCGA, there are six tumours with *BARD1* amplification and two with missense mutations of unknown significance. Given the lack of *BARD1* loss of function mutations identified in the TCGA ovarian cancer cohort, our results indicate that *BARD1* may play a more significant role in the development of ovarian cancer than previously recognized. Germline *BARD1* variants appear to convey a low-moderate risk of ovarian cancer, but do not appear to explain non-*BRCA1/2* breast and ovarian cancer heritability [18]. Molecular biology studies have demonstrated that *BARD1* can affect cell growth and tumour suppressor activity and play a role in neoplasia.

The extended sequencing panel identified genes that may have relevance for the identification of hereditary cancer syndromes. These gene mutations, if germline, would allow cascade testing of patient relatives. This, in turn, would allow potential risk reducing strategies to be employed, potentially preventing cancer in the affected relatives. Furthermore, inherited gene mutations can lead to consideration of other treatment strategies if a subsequent tumour develops.

There are several limitations to this study. First, targeted therapies, particularly those targeting tumours with deficiencies in homologous recombination, may be effective in more tumours than those identified here. There is ongoing research investigating the best and most appropriate biomarker for therapy response. Assessing tumour HRD can use methods, such as loss of heterozygosity, telomeric allelic imbalance, and large-scale state transitions, to name a few [35]. Therefore, the method used in this study may miss some HRD tumours. Another limitation is the lack of fresh frozen tissue on all of the cases. Finally, a more full-scale genomic sequencing may have identified additional, unexpected, therapeutic targets. An important question for this study is potential validation. Ideally, these results would be validated using an independent cohort. It would be interesting to validate and examine the variation in germline and somatic mutations in various populations, and this will require large scale international collaborations.

This study demonstrates that prospective sequencing of ovarian cancer leads to the identification of clinically and diagnostically relevant information. Ovarian cancer is a rare disease that should be considered in the development and deployment of personalized medicine. Approaches such as basket trials should be considered to allow patient management and therapy to be individually tailored. Beyond therapeutically actionable gene mutations, this study also identifies the relevance of prospective tumour sequencing for the identification of patients suffering from unidentified hereditary syndromes.

## 5. Conclusions

Ovarian cancer is a heterogeneous, rare, and aggressive malignancy that is ripe for precision-medicine-based therapeutic approaches. NGS based sequencing revealed actionable targets in most cases, including both potential therapeutics, as well as identification of hereditary cancer syndromes.

## Figures and Tables

**Figure 1 cancers-13-05240-f001:**
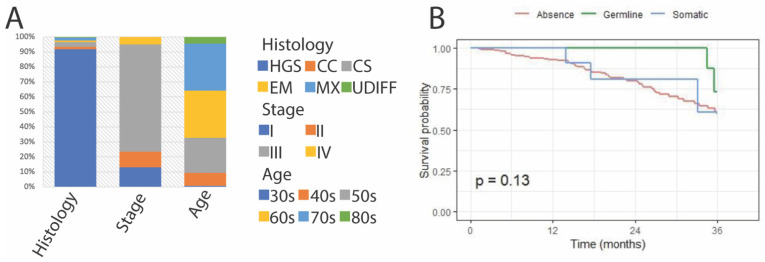
(**A**) Characteristics of the Karolinska cohort by histology, stage, and age. (**B**) Overall survival of the cohort by the BRCA1/2 pathogenic variant status (absent, germline or somatic).

**Figure 2 cancers-13-05240-f002:**
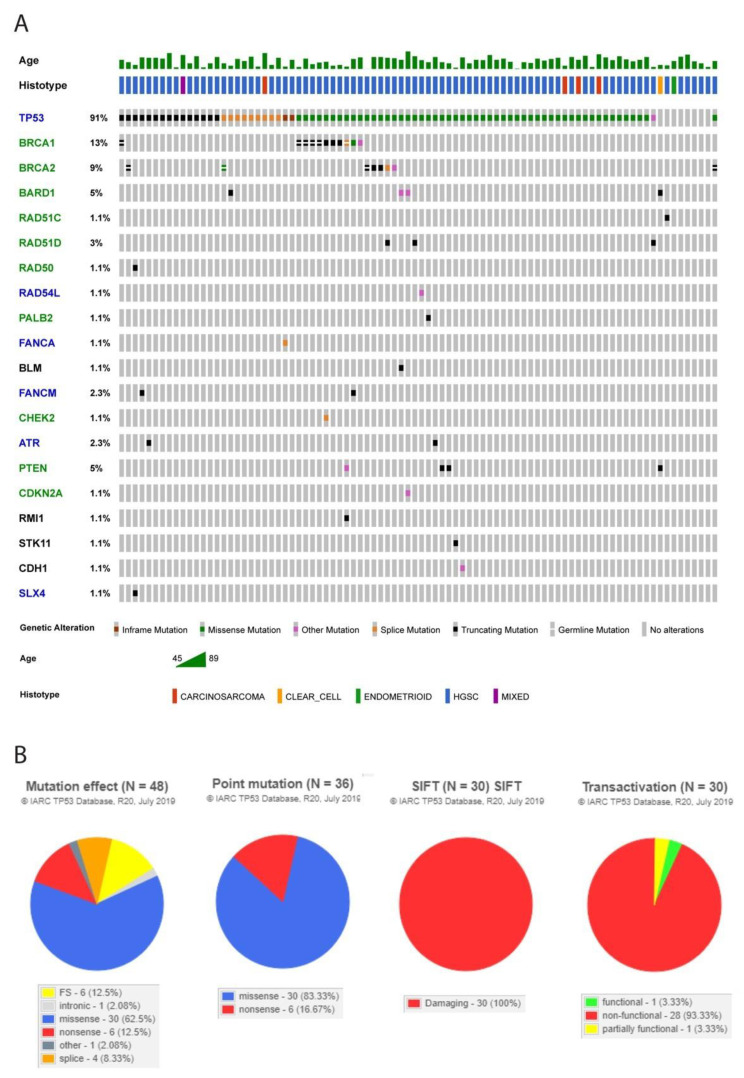
(**A**) Oncoprint of genomic alterations in the cohort. The text colour of the gene reflects the availability of targeted therapy, with the green text indicating an FDA approved drug, blue text indicating a drug in clinical trials, and black text indicating a gene related to an hereditary syndrome. (**B**) Detailed characteristics of the *TP53* loss of function mutations identified in the cohort. From left to right, these pie charts indicate (1) the mutation effect, (2) whether the point mutations are missense or nonsense, (3) whether the mutation was damaging by the SIFT analysis, and (4) the effect of the mutation on transactivation. These parameters may serve as markers of response to adavosertib, as discussed in the text.

**Figure 3 cancers-13-05240-f003:**
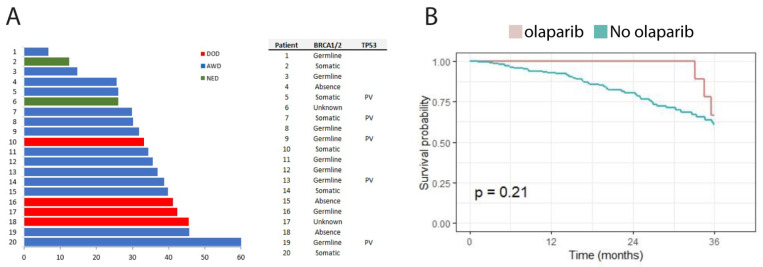
(**A**) Clinical follow-up of the 20 patients treated with olaparib, and their BRCA1/2 mutation status as germline, somatic or unknown. (**B**) Overall survival of the entire cohort grouped by those that received olaparib and those that did not.

**Table 1 cancers-13-05240-t001:** Clinicopathologic characteristics of the study cohort, as related to the detected *BRCA1/2* status. The clinicopathologic characteristics of the study cohort, for all of the patients, as well as for the patients with somatic, germline or no (i.e., “absent”) *BRCA1/2* variants. Note that 14 patients with tumour detected variants did not undergo follow-up germline testing and are not included in this table. The p-value indicates that when considering a level of statistical significance of 5%, there is evidence to say that the groups absence, germline, and somatic differ in age.

Variables	Total (260; 100.0%)	Absent (215; 82.7%)	Germline (29; 11.1%)	Somatic (16; 6.2%)	*p*-Value
**Age. years (median)**	66 (34–89)	67 (34–89)	59 (45–80)	65 (47–75)	0.001
**Histotype**					0.95
HGSC	239 (91.9%)	195 (90.7%)	28 (96.6%)	16 (100.0%)	
Clear cell carcinoma	4 (1.5%)	4 (1.9%)	0 (0.0%)	0 (0.0%)	
Carcinosarcoma	8 (3.1%)	8 (3.7%)	0 (0.0%)	0 (0.0%)	
Endometrioid carcinoma	3 (1.2%)	3 (1.4%)	0 (0.0%)	0 (0.0%)	
Mixed	5 (1.9%)	4 (1.8%)	1 (3.4%)	0 (0.0%)	
Undifferentiated carcinoma	1 (0.4%)	1 (0.5%)	0 (0.0%)	0 (0.0%)	
**Stage**					0.80
I	24 (9.2%)	21 (9.8%)	2 (6.9%)	1 (6.3%)	
II	19 (7.3%)	16 (7.4%)	3 (10.3%)	0 (0.0%)	
III	130 (50.0%)	105 (48.8%)	17 (58.6%)	8 (50.0%)	
IV	87 (33.5%)	73 (34.0%)	7 (24.2%)	7 (43.8%)	
**Fresh material**					0.04
Yes	83 (31.9%)	62 (28.8%)	12 (41.4%)	9 (56.3%)	
No	177 (68.1%)	153 (71.2%)	17 (58.6%)	7 (43.8%)	

**Table 2 cancers-13-05240-t002:** Pathogenic BRCA1 and BRCA2 variants detected in this study.

Germline or Somatic	RefSeq	Olaparib Therapy	BRCA	Variant cDNA	Protein Change	Other Names	Variant Type	Impact	ACGM Classification
Germline	NM_007294.3	YES	BRCA1	c.5153-1G>C		IVS18-1G>C	splice acceptor	Pathogenic	5
Germline	NM_007294.3	YES	BRCA1	c.181T>G	p.(Cys61Gly)	C61G	missense	Pathogenic	5
Germline	NM_007294.3	YES	BRCA1	c.2475del	p.(Asp825GlufsTer21)	2594delC	frameshift	Pathogenic	5
Germline	NM_007294.3		BRCA1	c.5096G>A	p.(Arg1699Gln)	5215G>A	missense	Pathogenic	5
Germline	NM_007294.3	YES	BRCA1	c.1772del	p.(Ile591LysfsTer8)	1891delT	frameshift	Pathogenic	5
Germline	NM_007294.3	YES	BRCA1	c.5095C>T	p.(Arg1699Trp)	5214C>T	missense	Pathogenic	5
Germline	NM_007294.3		BRCA1	c.4075C>T	p.(Gln1359Ter)	Q1312 *	nonsense	Pathogenic	5
Germline	NM_007294.3		BRCA1	c.3759dup	p.(Lys1254Ter)	3878insT	frameshift	Pathogenic	5
Germline	NM_007294.3		BRCA1	c.930del	p.(Gln310HisfsTer4)	1049delG	frameshift	Pathogenic	5
Germline	NM_007294.3		BRCA1	c.3048_3052dup	p.(Asn1018MetfsTer8)	3166ins5	frameshift	Pathogenic	5
Germline	NM_007294.3		BRCA1	c.1961dup	p.(Tyr655ValfsTer18)	2080insA	frameshift	Pathogenic	5
Germline	NM_007294.3		BRCA1	c.5266dup	p.(Gln1756ProfsTer74)	5382insC	frameshift	Pathogenic	5
Germline	NM_007294.3		BRCA1	c.3048_3052dup	p.(Asn1018MetfsTer8)	3166insTGAGA	frameshift	Pathogenic	5
Germline	NM_007294.3		BRCA1	c.3048_3052dup	p.(Asn1018MetfsTer8)	3166ins5	frameshift	Pathogenic	5
Germline	NM_007294.3	YES	BRCA1	c.3048_3052dup	p.(Asn1018MetfsTer8)	3166ins5	frameshift	Pathogenic	5
Germline	NM_007294.3		BRCA1	c.5444G>A	p.(Trp1815Ter)	5563G>A	nonsense	Pathogenic	5
Somatic	NM_007294.3	YES	BRCA1	c.2923C>T	p.(Gln975Ter)	Q928 *	nonsense	Pathogenic	5
Somatic	NM_007294.3	YES	BRCA1	c.3635C>G	p.(Ser1212Ter)	S1165 *	nonsense	Pathogenic	5
Somatic	NM_007294.3	YES	BRCA1	c.4186C>T	p.(Gln1396Ter)	Q1396 *	nonsense	Pathogenic	5
Somatic	NM_007294.3		BRCA1	c.3013del	p.(Glu1005AsnfsTer19)	3132delG	frameshift	Pathogenic	5
Somatic	NM_007294.3		BRCA1	c.1588del	p.(Glu530LysfsTer2)		frameshift	Pathogenic	5
Somatic	NM_007294.3	YES	BRCA1	c.3329del	p.(Lys1110SerfsTer7)	3448delA-ter1116	frameshift	Pathogenic	5
Somatic	NM_007294.3		BRCA1	c.505C>T	p.(Gln169Ter)	Q122 *	nonsense	Pathogenic	5
Somatic	NM_007294.3		BRCA1	c.5073A>T	p.(Thr1691=)		synonymous	Pathogenic	5
Somatic	NM_007294.3		BRCA1	c.4162C>T	p.(Gln1388Ter)	Q1341 *	nonsense	Pathogenic	5
Somatic	NM_007294.3		BRCA1	[GRCh37] t(17;1)(q21.31;q32.1)(41234522_41234523)			translocation	Pathogenic	5
Unknown	NM_007294.3	olaparib	BRCA1	c.848T>A	p.(Leu283Ter)		nonsense	Pathogenic	5
Unknown	NM_007294.3	olaparib	BRCA1	c.1066C>T	p.(Gln356Ter)	Q309 *	nonsense	Pathogenic	5
Unknown	NM_007294.3		BRCA1	c.1360_1361del	p.(Ser454Ter)	1479delAG	nonsense	Pathogenic	5
Germline	NM_000059.3		BRCA2	c.9097_9098insT	p.(Thr3033AsnfsTer11)	T3033fs	frameshift	Pathogenic	5
Germline	NM_000059.3		BRCA2	c.5946del	p.(Ser1982ArgfsTer22)	6174delT	frameshift	Pathogenic	5
Germline	NM_000059.3	olaparib	BRCA2	c.5851_5854del	p.(Ser1951TrpfsTer11)	6076del4	frameshift	Pathogenic	5
Germline	NM_000059.3		BRCA2	c.7976G>A	p.(Arg2659Lys)	8204G>A	missense	Pathogenic	5
Germline	NM_000059.3	olaparib	BRCA2	c.3103G>T	p.(Glu1035Ter)	3331G>T	nonsense	Pathogenic	5
Germline	NM_000059.3		BRCA2	c.5146_5149del	p.(Tyr1716LysfsTer8)	5373delGTAT	frameshift	Pathogenic	5
Germline	NM_000059.3		BRCA2	c.5576_5579del	p.(Ile1859LysfsTer3)	5804_5807delTTAA	frameshift	Pathogenic	5
Germline	NM_000059.3	olaparib	BRCA2	c.5946del	p.(Ser1982ArgfsTer22)	6174delT	frameshift	Pathogenic	5
Germline	NM_000059.3		BRCA2	c.4677del	p.(Phe1559LeufsTer9)		frameshift	Pathogenic	5
Germline	NM_000059.3		BRCA2	c.956dup	p.(Asn319LysfsTer8)	1184insA	frameshift	Pathogenic	5
Germline	NM_000059.3		BRCA2	c.5754_5755del	p.(His1918GlnfsTer5)		frameshift	Pathogenic	5
Germline	NM_000059.3		BRCA2	c.3847_3848del	p.(Val1283LysfsTer2)	4075_4076delGT	frameshift	Pathogenic	5
Somatic	NM_000059.3		BRCA2	[GRCh37] t(13;6)(q13.1;q14.3)(32890451_32890530)			translocation	Pathogenic	5
Somatic	NM_000059.3	olaparib	BRCA2	c.631 + 1G>A		IVS7+1G>A	splice donor	Pathogenic	5
Somatic	NM_000059.3		BRCA2	c.7133C>G	p.(Ser2378Ter)	7361C>G	nonsense	Pathogenic	5
Somatic	NM_000059.3	olaparib	BRCA2	c.2539A>T	p.(Arg847Ter)	2767A>T	nonsense	Pathogenic	5
Somatic	NM_000059.3		BRCA2	c.5351dup	p.(Asn1784LysfsTer3)	5579insA	frameshift	Pathogenic	5
Unknown	NM_000059.3		BRCA2	c.161del	p.(Asn54ThrfsTer26)	0389delA-ter79	frameshift	Pathogenic	5
Unknown	NM_000059.3		BRCA2	c.8021dup	p.(Ile2675AspfsTer6)	8249insA	frameshift	Pathogenic	5
Unknown	NM_000059.3		BRCA2	c.9253dup	p.(Thr3085AsnfsTer26)		frameshift	Pathogenic	5
Unknown	NM_000059.3		BRCA2	c.6096del	p.(Ile2033Tyrfs *7)		frameshift	Pathogenic	5

* stop codon.

## Data Availability

The data presented in this study are available on request from the corresponding author. The data are not publicly available due to ethical considerations.

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
