# Peer review of "Precision Oncology of High-Grade Ovarian Cancer Defined through Targeted Sequencing"

_cancers, 2021, doi:10.3390/cancers13205240_

Round 1
Reviewer 1 Report
Review report
The authors collected 185 FFPE and 89 fresh frozen samples from patients with high-grade ovarian cancers and performed sequencing on BRCA1 and BRCA2 genes. In addition, they carried out targeted enrichment of DNA from the 89 fresh frozen samples and had a targeted sequencing on a panel of 55 other genes. Based on these results, they selected 20 patients, who received PARPi treatment.
The works resulted important information on high-grade ovarian cancers and should be provided to the scientific society.
Nevertheless, the manuscript is not well constructed to give the readers an overview and thus needs to be revised substantially.
In details:
Title:
- You just sequenced BRCA1/2 and partly a targeted gene sets. You cannot call this genomic landscape. Please precise your results.
- What is a clinical sequencing?
Introduction:
Line 49: a world wild incidence should be given instead of the one only in UK;
Methods:
- Gene name is always written italic.
- It is very important to indicate the estimation of tumor cell proportion and the allele rates of the detected mutations in all samples. I suggest adding a supplementary table providing this information of all samples.
- The authors mentioned that a total of 274 tumors were sequenced. However, table one only show a total of 260. Where are the missing 14?
- Table 1. What does the p-value indicate?
- Have you, at lease in selected samples, validated the detected mutations using other techniques, for example, Sanger sequencing?
- The authors used “pathogenic / likely pathogenic variants“ in several places. However, there was not a definition on this.
- Line 132/133: how is the percentage of TP53 mutation in HGS epithelial ovarian cancer??
- The selection of patients for PARP inhibitor treatment is missing in the methods. Please provide a table indicating the selection criteria.
- Most of the sequencing data showed equal mutations of BRCA1/2 mutations, some find less BRCA1 What is your comment on this?
Author Response
Thank you for your comments. Here is our point by point response and revision.
Title:
1. You just sequenced BRCA1/2 and partly a targeted gene sets. You cannot call this genomic landscape. Please precise your results.
Thank you for this comment. We have adjusted the title to be more precise.
2. What is a clinical sequencing?
“Clinical sequencing” is prospective sequencing of patient materials at or near the time of diagnosis in a clinically licensed lab, with the goal of deriving molecular information to be used in the patients therapy. In this case, sequencing for BRCA1 and BRCA2 was used to determine whether the patients were eligible for PARP inhibitor therapy. We have added a clarification to the methods regarding this term.
Introduction:
Line 49: a world wild incidence should be given instead of the one only in UK;
We have changed this to reflect worldwide deaths, and modified the citation to reflect this.
Methods:
1. Gene name is always written italic.
We have corrected this.
2. It is very important to indicate the estimation of tumor cell proportion and the allele rates of the detected mutations in all samples. I suggest adding a supplementary table providing this information of all samples.
Thank you for this comment. Yes, allele rates would be very interesting to try to calculate. Unfortunately, we have quality control parameters that dictate a minimum required tumor cell proportion (>30%) and a minimum allele rate of >5%, but we do not have more detailed information case by case. We have added this information to the methods.
3. The authors mentioned that a total of 274 tumors were sequenced. However, table one only show a total of 260. Where are the missing 14?
These 14 patients had not been seen at the genetic counseling clinic and/or their germline sequencing was not completed at the time of data freeze for this study. Thus, we do not know their hereditary status. We have double checked that this is mentioned as part of the Table legend.
4. Table 1. What does the p-value indicate?
The p-value indicates that when considering a level of statistical significance of 5%, there is evidence to say that the groups absence, germline and somatic differ in age.
5. Have you, at lease in selected samples, validated the detected mutations using other techniques, for example, Sanger sequencing?
Yes, we have validated the detected mutants in selected samples. We have added text describing this to the materials and methods.
“A majority of the reported BRCA1 and BRCA2 variants were confirmed using Sanger sequencing (70%), smMIPs (9.4%, only variants detected by hybrid capture) or pyrosequencing (9.4%), with 100% success rate. A minority of TP53 variants were also confirmed by Sanger sequencing (2 variants) or smMIPs (11 variants) with 100% success rate. Variants in other genes were reviewed in IGV.”
6. The authors used “pathogenic / likely pathogenic variants“ in several places. However, there was not a definition on this.
ClinVar type 5 is Pathogenic, Type 4 is Likely Pathogenic. We have added text to the methods specifying this.
7. Line 132/133: how is the percentage of TP53 mutation in HGS epithelial ovarian cancer??
The TCGA identified 88% of high grade serous ovarian cancers as having a TP53 mutation. We have rewritten this sentence – it mistakenly said the search was done in the PanCancer Atlas.
8. The selection of patients for PARP inhibitor treatment is missing in the methods. Please provide a table indicating the selection criteria.
We have added this text to the Methods. Since the indication for PARPi therapy was rather straightforward (BRCAm disease (either sBRCA or gBRCA) with response to >/= 2nd line platinum based chemotherapy) we feel it works best as an additional sentence, and not as a table.
9. Most of the sequencing data showed equal mutations of BRCA1/2 mutations, some find less BRCA1 What is your comment on this?
Estimating the expected quantity of BRCA1 and BRCA2 mutation in a given population is difficult for several reasons. First, this study included only women with unknown germline status. Clearly, different countries and regions have different systems for identification of BRCA1/2 carriers. Second, this study is focused on women presenting with ovarian cancer. How this relates to the incidence of BRCA1/2 germline carriers in women who have not yet developed ovarian cancer is unclear, and can also depend on risk reducing salpingo-oophorectomy availability. Third, the incidence of somatic BRCA1/2 variants may also depend on environmental and patient factors. Thus, the expected quantity of mutations may vary, perhaps widely, between studies.
Reviewer 2 Report
The article entitled "The Genomic Landscape of High-grade Ovarian Cancer Defined Through Clinical Sequencing" by Wessman et. al. screens 260 ovarian cancer patients for mutations in genes relating to the homologous recombination pathway as well as a number of other driver mutations such as TP53 and PTEN. The authors found 21.5% of their cohort had BRCA1/2 mutations and four patients with BARD1 mutations. Of the patients sequenced for additional driver mutations, 90% had a mutation in TP53.
The rates of BRCA1/2 mutations reported in this manuscript are in line with those previously reported in larger cohorts, as are the rates of other well-established driver genes such as TP53, PTEN and ATR. As this study is aiming to show the clinical utility of sequencing ovarian cancer tumours in the hopes that the patient will be matched to a targeted therapy, it may be useful to include a table/figure/schematic showing the proportion of patients within the study that had a mutation matched to an FDA-approved/ovarian cancer available drug, then those that have mutations matched to a drug that is currently in clinical trials etc. The data in figure 3 could then be used as an example of the benefits of matching mutations to targeted therapy to really drive home this point. One question regarding Figure 3b - is the survival data of the entire cohort or just the 50 patients with mutations in BRCA1/2? It is unclear from the Figure legend and associated text.
In figure 2b, the interpretation of results is difficult. It is not immediately obvious that one pie chart flows into the next and so on without reading the associated text in the results section. The figure needs to be amended or extra detail needs to be added to the figure legend.
On a technical point the age of the patients should be displayed at age at diagnosis, rather than just age. Also, all decimal points in the manuscript appear to be represented as commas. This should be corrected.
Finally, an important aspect of personalised therapy is the type of molecular analysis being performed. There are many considerations to this eg. tissue availability/quality, analysis depth, cost etc. which should be explored as part of the discussion. Of particular importance for this study is that only BRCA1/2 mutations and the commonly mutation HR pathway genes are explored. It is now well known that there are multiple factors that contribute to BRCA'ness' and whether a tumour will respond to PARPi (BRCA1/RAD51C promoter methylation status, LOH signatures etc). Addressing this will provide more depth to the discussion and overall conclusions.
Author Response
Thank you for your helpful comments. These are marked below with a bullet point, followed by our response.
- The rates of BRCA1/2 mutations reported in this manuscript are in line with those previously reported in larger cohorts, as are the rates of other well-established driver genes such as TP53, PTEN and ATR. As this study is aiming to show the clinical utility of sequencing ovarian cancer tumours in the hopes that the patient will be matched to a targeted therapy, it may be useful to include a table/figure/schematic showing the proportion of patients within the study that had a mutation matched to an FDA-approved/ovarian cancer available drug, then those that have mutations matched to a drug that is currently in clinical trials etc. The data in figure 3 could then be used as an example of the benefits of matching mutations to targeted therapy to really drive home this point.
This is an excellent suggestion. We have revised Figure 2 to include information about the availability of 1) FDA-approved available drug, 2) drug in clinical trials, and 3) mutation related to a hereditary syndrome.
- One question regarding Figure 3b - is the survival data of the entire cohort or just the 50 patients with mutations in BRCA1/2? It is unclear from the Figure legend and associated text.
This figure is showing the survival of patients that received Olaparib versus those that did not, and includes the entire cohort. We have adjusted the Figure text to clarify this.
- In figure 2b, the interpretation of results is difficult. It is not immediately obvious that one pie chart flows into the next and so on without reading the associated text in the results section. The figure needs to be amended or extra detail needs to be added to the figure legend.
Thank you for this comment. We have added details to legend for figure 2b to attempt to fic this issue.
- On a technical point the age of the patients should be displayed at age at diagnosis, rather than just age. Also, all decimal points in the manuscript appear to be represented as commas. This should be corrected.
Yes, this is the age we have used. We have clarified this in the methods. We have also corrected the use of commas instead of decimal points.
- Finally, an important aspect of personalised therapy is the type of molecular analysis being performed. There are many considerations to this eg. tissue availability/quality, analysis depth, cost etc. which should be explored as part of the discussion. Of particular importance for this study is that only BRCA1/2 mutations and the commonly mutation HR pathway genes are explored. It is now well known that there are multiple factors that contribute to BRCA'ness' and whether a tumour will respond to PARPi (BRCA1/RAD51C promoter methylation status, LOH signatures etc). Addressing this will provide more depth to the discussion and overall conclusions.
We have added text to the discussion, including a section on limitations that includes some overview of HRD detection.
Reviewer 3 Report
Thank you for your submission, which is further evidence of the utility of genomic screening in the clinical management of ovarian cancer.
Some queries to be addressed in a revision:
In the short summary, it is reported you have identified 59 pathogenic mutations in BRCA1/2. Further down in the manuscript, it is detailed that in fact it was 50 pathogenic mutations that have been identified in the BRCA genes, and 9 variants of unknown significance (VUS). Please revise.
Consistently in the report, within a gene variants classified as being as pathogenic (or likely pathogenic) are being combined with VUS as a "total actionable mutations". For example, in the oncoprint - 4 of the categories of mutations are described as being "of unknown significance". In all aspects of this study, it would serve better to clearly define those mutations that are of C4/5 class from those that are C3, and keep them separate. In particular because the clinical applicability of a VUS (in both a familial cancer setting, or for the purposes of drug eligibility) is not equivalent.
It is unclear how germline or somatic status has been determined in the patient set. It is described that the status was undetermined for 14 tumors, as patients did not undergo follow up germline testing - but there is no description of how that follow up testing occurred. Presumably through the described referral for genetic counselling - but why the 94/274 specifically were referred only is not explained.
Author Response
Thank you for your comments. We have attempted to address them as best we can. Your comments are marked with a bullet point, followed by our response.
- In the short summary, it is reported you have identified 59 pathogenic mutations in BRCA1/2. Further down in the manuscript, it is detailed that in fact it was 50 pathogenic mutations that have been identified in the BRCA genes, and 9 variants of unknown significance (VUS). Please revise.
The correct number is 50, and we have edited the text to reflect this.
- Consistently in the report, within a gene variants classified as being as pathogenic (or likely pathogenic) are being combined with VUS as a "total actionable mutations". For example, in the oncoprint - 4 of the categories of mutations are described as being "of unknown significance". In all aspects of this study, it would serve better to clearly define those mutations that are of C4/5 class from those that are C3, and keep them separate. In particular because the clinical applicability of a VUS (in both a familial cancer setting, or for the purposes of drug eligibility) is not equivalent.
Yes, this is an important point. The designation in the oncoprint was in error based upon an error in the entry of the data to the oncoprint software. All the mutations discussed as actionable and presented in the manusceript and the oncoprint are pathogenic. The only VUS identified and described are for BRCA1 and BRCA2 and they are only presented in the supplementary table. Thank you for identifying this.
- It is unclear how germline or somatic status has been determined in the patient set. It is described that the status was undetermined for 14 tumors, as patients did not undergo follow up germline testing - but there is no description of how that follow up testing occurred. Presumably through the described referral for genetic counselling - but why the 94/274 specifically were referred only is not explained.
We have added text to the methods to describe this. Fourteen patients had not been evaluated by clinical genetics at the time the data for this study was locked. At this time, in Sweden, only selected patients were referred to clinical genetics. This is in contrast to many countries where all women with ovarian cancer were referred. According to guidelines this was to be done in cases with diagnosis at young age and/or a medical history of breast cancer and/or at least one first degree relative with breast- or ovarian cancer. After the implementation of sBRCA-screening at diagnosis, all patients with a detected sBRCA-mutation were referred to Clinical Genetics. Importantly, cases with a previously known gBRCA mutation were of course not referred.
Reviewer 4 Report
Wessman et al. studied the molecular profile of a cohort of high grade ovarian carcinoma by performing targeted next generation sequencing to identify potential targets for precision therapy. High grade ovarian carcinoma is one of the commoner forms of gynecological malignancy and its known for its adverse prognosis. Identification of therapeutic targets is crucial to expand the therapeutic options and this study takes an important step in that direction. The manuscript is well-written and the conclusion is supported by the results. Detection of a subset of tumors with co-existing mutations in BARD1 and TP53 or PTEN was particularly interesting.
Specific points:
- Abstract: The authors need to clarify that next generation sequencing using a targeted panel was used for this study.
- Materials and methods: The authors mention that histotype and the grade of the included tumors were established using standard criteria. Was this based only on histology or were immunohistochemical markers also used? Which immunohistochemical markers were used and how were they interpreted? Were there any diagnostically challenging cases in this cohort with discordance in the histological appearance and the immunohistochemistry results? How were the diagnoses established in such cases? Was the mismatch repair status determined by using a microsatellite instability panel or methylation sequencing? These are valuable information for the manuscript and should be provided. If these ancillary investigations were not performed, the authors should address this in the discussion as a limitation of the study.
- The authors mention that fresh frozen tissue was sequenced for 89 cases and FFPE tissue for 185 cases. Was fresh frozen tissue not obtained for every case and for how many cases both fresh frozen tissue and FFPE tissue were available?
- The complete list of the amplicons used in the sequencing panel along with the coverage should be provided in the supplementary materials.
- What was the rationale of performing the extended sequencing on 81 of the high grade serous ovarian carcinomas? How was this number (n = 81) determined?
- Did the authors analyze copy number alterations in the included tumors?
- The authors detected a number of germline BRCA mutations in the selected tumors. Did these women have any relevant familial history?
- The authors could present a graph comparing the frequencies of the mutations detected in the reported cohort and the TCGA cohort.
- The authors need to discuss the limitations of the study in the discussion. They could also discuss what are the potential steps for validation of the findings.
Author Response
Thank you for your helpful comments. We believe we have addressed them, to the benefit of the manuscript. Your comment is marked with a bullet point, followed by our response.
Specific points:
- Abstract: The authors need to clarify that next generation sequencing using a targeted panel was used for this study.
We have clarified this.
- Materials and methods: The authors mention that histotype and the grade of the included tumors were established using standard criteria. Was this based only on histology or were immunohistochemical markers also used? Which immunohistochemical markers were used and how were they interpreted? Were there any diagnostically challenging cases in this cohort with discordance in the histological appearance and the immunohistochemistry results? How were the diagnoses established in such cases? Was the mismatch repair status determined by using a microsatellite instability panel or methylation sequencing? These are valuable information for the manuscript and should be provided. If these ancillary investigations were not performed, the authors should address this in the discussion as a limitation of the study.
This is a very interesting point, and we have added text to the methods to clarify this. Briefly, both histology and immunohistochemistry were used to diagnose the tumors. The immunohistochemical panel typically included PAX8, CK7, CK20, WT1, ER, PR, and P53. Mismatch repair status was not routinely determined for these tumors (in contrast to clear cell or endometrioid FIGO 1 and FIGO 2). A detailed histological characterization of these tumors seems beyond the scope of this manuscript, but in our lab most high grade serous carcinomas are readily recognizable on H&E and have the correct, characteristic IHC profile.
- The authors mention that fresh frozen tissue was sequenced for 89 cases and FFPE tissue for 185 cases. Was fresh frozen tissue not obtained for every case and for how many cases both fresh frozen tissue and FFPE tissue were available?
Cases were either sequenced using fresh tissue (if available) or FFPE, but not both. This is because fresh tissue was not obtained from every case. Although FFPE material is available on all cases, sequencing was only performed on one or the other (fresh or FFPE). We have clarified this in the results.
- The complete list of the amplicons used in the sequencing panel along with the coverage should be provided in the supplementary materials.
We are providing this as supplemental data.
- What was the rationale of performing the extended sequencing on 81 of the high grade serous ovarian carcinomas? How was this number (n = 81) determined?
We could only perform the extended panel on tumors with fresh frozen material due to the pipeline that was available.
- Did the authors analyze copy number alterations in the included tumors?
Copy number alterations were analyzed in the fresh frozen tumors using the software XHMM. We have clarified this in the text.
- The authors detected a number of germline BRCA mutations in the selected tumors. Did these women have any relevant familial history?
Yes, 16 of these women had a family history. We have added this to the results sections.
- The authors could present a graph comparing the frequencies of the mutations detected in the reported cohort and the TCGA cohort.
We have analysed this as a Table, and it is found as Supplementary Table 1.
- The authors need to discuss the limitations of the study in the discussion. They could also discuss what are the potential steps for validation of the findings.
We have added a discussion of limitations and validation to the discussion.
Round 2
Reviewer 1 Report
The authors have improved the presentation of the data substantially. The manuscript is acceptable with minor text editing.
Author Response
- The authors have improved the presentation of the data substantially. The manuscript is acceptable with minor text editing.
Thank you. We have performed additional text edits.
Reviewer 3 Report
Thank you for your revisions.
Author Response
- Thank you for your revisions.
Thank you.
Reviewer 4 Report
In general, the authors have adequately addressed my comments.
A few minor points remain:
- The authors provide a definition of ‘clinical sequencing’ in the text of the manuscript; however, the meaning of clinical sequencing is not apparent by reading the title. The authors could consider changing this to ‘targeted sequencing’ or ‘next-generation sequencing’.
- Materials and Methods; Line 84: P53 should be p53
- Supplementary materials: The authors have provided the list of amplicons included in the panel. They need to provide the coverage of the individual genes as well.
Author Response
- In general, the authors have adequately addressed my comments.
- A few minor points remain:
- The authors provide a definition of ‘clinical sequencing’ in the text of the manuscript; however, the meaning of clinical sequencing is not apparent by reading the title. The authors could consider changing this to ‘targeted sequencing’ or ‘next-generation sequencing’.
We have changed the title to "targeted sequencing"
- Materials and Methods; Line 84: P53 should be p53
We have corrected this.
- Supplementary materials: The authors have provided the list of amplicons included in the panel. They need to provide the coverage of the individual genes as well.
We have provided Table S3 with details of the coverage of the individual genes. We hope this will be helpful.